# Food Waste Originated Material as an Alternative Substrate Used for the Cultivation of Oyster Mushroom (*Pleurotus ostreatus*): A Review

Ana Doroški [1,*], Anita Klaus [1], Anet Režek Jambrak [2] and Ilija Djekic [1,*]

1   Institute for Food Technology and Biochemistry, Faculty of Agriculture, University of Belgrade, Nemanjina 6, 11080 Belgrade, Serbia
2   Faculty of Food Technology and Biotechnology, University of Zagreb, 10000 Zagreb, Croatia
*   Correspondence: ana.doroski@agrif.bg.ac.rs (A.D.); idjekic@agrif.bg.ac.rs (I.D.)

**Abstract:** *Pleurotus ostreatus* (*P. ostreatus*) is considered a high-quality food, rich in proteins and bioactive compounds important for maintaining human health. Lately, a commonly used substrate for oyster mushroom cultivation—wheat straw, is more often replaced by alternative cellulose substrates originated from the agricultural and food industry. Utilization of wastes for mushroom cultivation has its added value: sustainable food waste management, production of high-quality food from low quality waste, as well as solving environmental, economic and global issues. This overview covered three categories of food waste: food-processing wastes, agro-cereal wastes and nut–fruit wastes, the most used for the cultivation *P. ostreatus* in the period of 2017–2022. Analyzed studies mostly covered the productivity and chemical characterization of the substrate before and after the cultivation process, as well as the morphological characteristics of the fruiting bodies cultivated on a specific substrate. Chemical analyses of mushrooms cultivated on food waste are not adequately covered, which gives room for additional research, considering the influence of substrate type and chemical quality on the fruiting bodies chemical composition.

**Keywords:** *Pleurotus ostreatus*; oyster mushroom; food waste; sustainability; cultivation; substrate

## 1. Introduction

Different mushroom species are consumed in many countries around the world as traditional and functional foods [1], as well as a delicacy, for their specific flavor and texture properties [2]. On the other hand, public and scientific interest in mushroom secondary metabolites and bioactive components with their antioxidant, antimicrobial, antitumor, antiviral and immunomodulatory properties are increasing [1,3,4].

According to FAOSTAT data, total world production of mushrooms in 2020 was 43 million metric tons, with a total of 1.3 million metric tons in Europe. The leading producer was China with a total of 40 million metric tons. The most produced mushroom in the world is *Agaricus bisporus* (button mushroom), followed by *Lentinula edodes* (shiitake) and *Pleurotus ostreatus* (oyster mushroom) [2].

In general, mushroom species are categorized in three groups, according to Kalač [2]. This categorization is based on mushroom nutritional strategy. The first group includes mycorrhizal or symbiotic species, which form mutually favorable connections with the host trees. The second group, named saprotrophic species or saprophites, derive their nutrients from dead organic material. This species is the basis for commercial cultivated production. The third group, parasitic species, lives in non-symbiotic relationship on the other species.

The scientific classification of *Pleurotus* species belongs to the Kingdom of *Fungi*, Division of *Basidiomycota*, Class of *Agaricomycetes*, Order of *Agaricales*, Family of *Pleurotaceae* and Genus of *Pleurotus*, defined by the German mycologist Paul Kummer in 1871 [5].

One of the most consumed mushroom species, *P. ostreatus*, belongs to the "white-rot fungi" category, which produces the lignolytic enzymes laccase and peroxidase and makes them able to degrade lignin, in addition to cellulose. With regard to this feature, this species may be cultivated on a wide range of agro-industrial, food and cellulose wastes, as replacements for ordinary substrates used in industrial production [3,6], which affirms the fact that oyster mushrooms, among the diverse white-rot fungi species, represent high-quality food originating from low quality waste [7].

Apart from its delicate taste and texture, *Pleurotus* Genus represents a nutritionally rich food with a high content of crude proteins and dietary fiber [5,8]. Many studies have focused on the simplicity of the *P. ostreatus* cultivation, which is attractive for scientific and commercial utilization [8]. The study of Rodriguez Estrada and Pecchia [8] paid close attention to the cultivation method of *P. ostreatus*, including spawn preparation, substrate manipulation, as well as alternative cultivation methods, harvesting and the possibility of disease during fructification and mycelium running through the substrate. The focus on waste biodegradation and enzymatic activity of *Pleurotus* species was the highlight of the study of Sekan et al. [9], distinguishing the green potential of this fungi from this perspective. A new research study examined enzyme production and mycelium growth of *Pleurotus* spp. improved by green illumination [10], adding value to the sustainability aspect.

An aspect of food waste utilization was defined by Morone et al. [11] as a sustainable food waste management strategy with the main goal to isolate high value waste material. Essentially, isolating any food waste material from the household or production process and using it as a raw material for a new production process that generates near zero-waste and makes an effective circular economy cycle is worthy of attention. Beneficial aspects of food waste recycling are diverse and include solving environmental [12,13], social [14] and economic [15] global issues.

In general, in addition to using waste as a resource for a new production process, a huge beneficial aspect of food waste recycling is the elimination of environmental pollution through the food chain [16].

The aim of this review was to examine the newest studies on the importance of food waste use for *P. ostreatus* cultivation during 2017–2022. Food wastes are presented in three groups: food-processing wastes, agro-cereal wastes and nut–fruit wastes, related to their industry origin. The most important methods regarding the contribution to the quality of produced mushrooms are discussed and covered within the topics.

## 2. Materials and Methods

### 2.1. Research Methodology

A review of 36 selected scientific papers covering food waste substrates mostly used for *P. ostreatus* cultivation in the last 5 years (2017–2022) is presented in Figure 1, divided into three food waste categories: food-processing wastes, agro-cereal wastes and nut–fruit wastes. An overview of keywords used in the studies related to the food waste substrate utilization for *P. ostreatus* cultivation was generated by the VOSviewer online program and is displayed in Figure 2. The review of the literature was performed using the main characteristics associated with *P. ostreatus* cultivation. The research criteria were applied through Web of Science search engine as follows: range of years (2017–2022); type of articles (research); keywords used ("*Pleurotus ostreatus*", "cultivation", "food waste", "substrate"). The selection of the first fifty papers listed in the Web of Science search engine was narrowed down by the food waste criteria mentioned above.

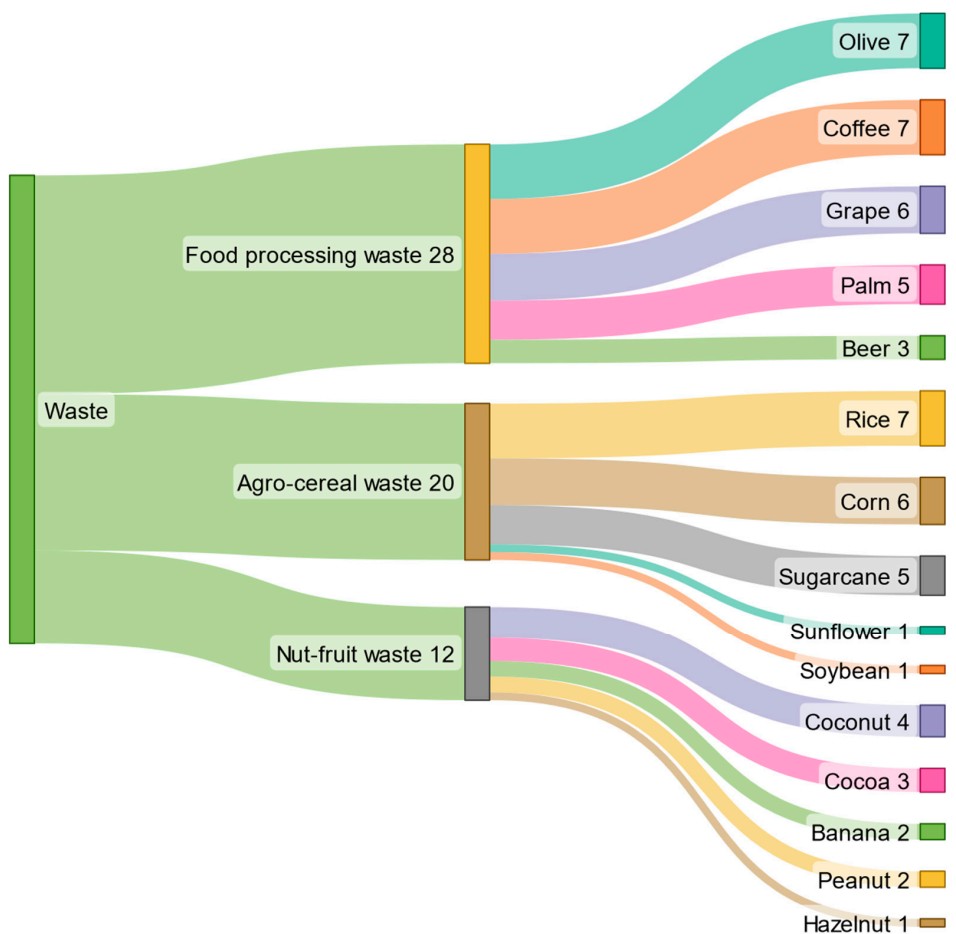

**Figure 1.** Sankey chart of the most used food-processing, agro-cereal and nut–fruit wastes for *Pleurotus ostreatus* cultivation.

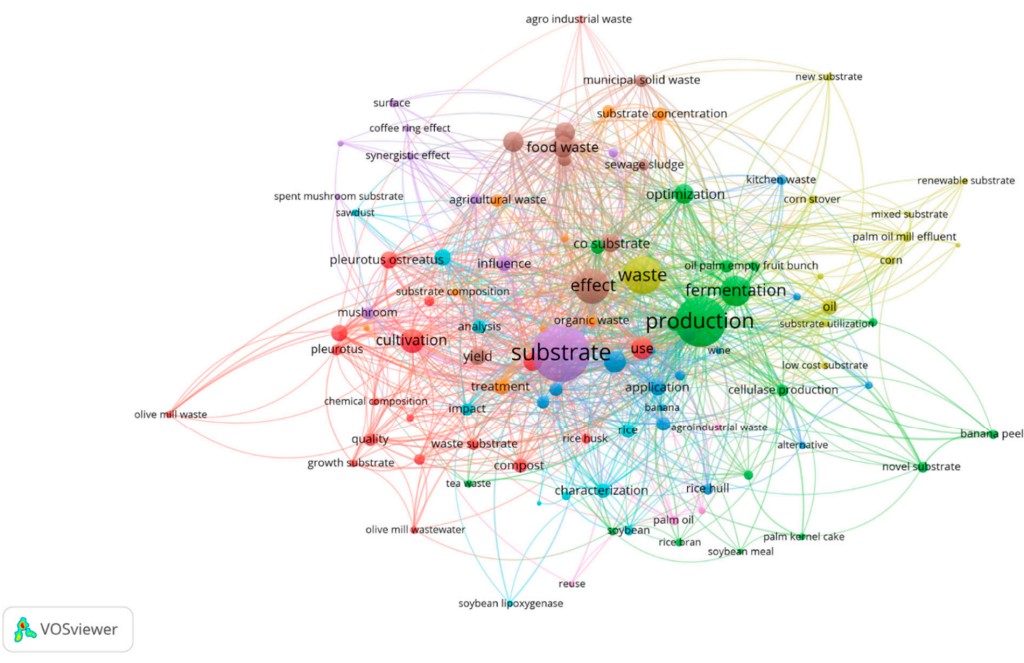

**Figure 2.** An overview of keywords used in the studies related to food waste substrate utilization for *Pleurotus ostreatus* cultivation: generated by VOSviewer.

### 2.2. Food-Processing, Agro-Cereal and Nut–Fruit Wastes Used for Pleurotus ostreatus Cultivation: An Overview

Regarding the aim of this research, Figures 1 and 2 were created in order to cover and discuss the newest evaluation and importance of food waste use for *P. ostreatus* cultivation. Namely, Figure 1 presents the most used wastes listed in the three categories previously mentioned and their frequency of occurrence in scientific papers published in 2017–2022 through a Sankey chart. The most frequent wastes belong to the food-processing wastes category, where olive oil and coffee wastes are covered the most. In the category of agro-cereal wastes, rice waste appears the most, followed by corn and sugarcane waste. Fruit–nut wastes are the least covered, but still with significant contribution.

An overview of keywords used in the studies related to food waste substrate utilization for *Pleurotus ostreatus* cultivation were generated by the VOSviewer online program and presented in Figure 2. This figure aims to introduce the frequency of keywords repeated through the scientific papers found via previously mentioned research criteria. The surface area of each point belonging to the mentioned figure represents the frequency value. Thus, the following keywords dominate: "substrate", "production", "waste", "effect", "fermentation", "cultivation" and "food waste", while keywords such as "olive mill wastewater", "palm oil", "soybean", "banana peel", "rice bran", etc., also occupy significant space in the research scope.

## 3. Results and Discussion

### 3.1. Potential of Food Waste

Accumulation of organic waste mainly originates from different food industry processes. It varies in composition, water content and pH value, carrying the risk of bacterial contamination and negative environmental influences [17,18]. Valorization of food wastes and the application of new technologies to eliminate waste by producing value-added products through bioconversion processes is the only way to manage this problem. That way, food waste may become renewable energy with high cellulose and lignin content when it is plant originated [18]. Various technologies have appeared to solve the waste generation problem: for example, transforming apple peel waste into ethanol and acetic acid by enzymatic hydrolysis of cellulose [19]. Banana leaf waste is a valuable raw material source, suitable for the production of lignocellulosic micro/nanofibers [20]. Additionally, food waste may bioconvert into other various products, such as biofuels, industrial enzymes, nutraceuticals or even biodegradable plastics. This technology reveals organic waste as a reservoir of other value-added chemicals and justifies the new concept "bioeconomy", the conversion of organic resources into cost-effective bioproducts and energy [18].

Food supply chain waste is divided to pre- and post-consumer waste, all originating from different food chain phases: food production, processing, distribution and consumption [17]. Agricultural and food-processing waste is the most significant for mushroom cultivation process.

While growing on the olive mill solid waste, *P. ostreatus* utilizes the lipid-soluble ingredients, which shift the degradation of the structural carbohydrates and could affect the glucan concentration of the mushrooms. Olive mill waste is also a strong source of nitrogen, minerals and other nutrients [21]. β-glucans are structural components of the mushroom cell wall [22]. In contrast to β-glucans, there are limited reports of α-glucans structure, due to their low concentration in mushrooms and their uncertain biological function [21].

According to Reverberi et al. [23], cultivation of *P. ostreatus* on olive mill wastes stimulates the β-1,3-glucan synthase activity of *P. ostreatus*, whose mechanism is based on an oxidative stress response by β-glucan biosynthesis. The source of oxidative stress may be the presence of oxidizing phenol compounds or the reactive oxygen species. The potential hindrance of using enzymatic methods for glucan content determination in mushroom-fruiting bodies is the occurrence of fiber residues, such as chitin-glucan complex, which is

an insoluble material that limits the diffusion of enzymes during β-glucan determination and results in lower β-glucan content measurements [22].

As olive mill waste is a hazardous waste, a huge benefit of using it as mushroom cultivation substrate is degradation and utilization while producing pharmacologically active compounds [21]. Parameters such as earliness and mycelial growth rates showed the advantage to composted olive mill waste with regard to raw treatment and control sample as well. Phytotoxicity tests showed the effectiveness of composting treatment. Composting treatment may provide certain trigger compounds, which are able to magnify the lignin degrading system of white-rot fungi [24]. Thus, composted olive mill waste showed significantly higher pH (7.5–8.0) compared to the acceptable acidic conditions favorable for mycelial growth, although it showed high yields and efficient mycelial growth comparable to raw olive mill waste and wheat straw. The growing values of a two-phase olive mill waste contribution in wheat straw substrate, despite high phenolic compound values, affected the overall biotoxicity of the substrate as well. It was noticeable by the negative impact on mushroom yields when the supplementation of raw olive mill waste reached 60% volume.

The main advantages of coffee pulp as an alternative substrate for *Pleurotus* spp. cultivation is that it is a nutritive substrate for mushroom growth, available as a no-cost waste product and can could be dehydrated and stored for a long time with no side effects on mushroom yield. It is also seasonally available as a fresh material from the coffee production. Additionally, coffee pulp degraded by mushrooms could be considered as a valuable post-product used as organic fertilizer and vermicompost [25].

### 3.2. Research Covering Substrates

### 3.2.1. *P. ostreatus*: Substrate Preparation

The substrate degradation capacity of *Pleurotus* spp. is related to its ability to secrete specific enzymes, such as laccases, cellulases, hemicellulases, peroxidases and xylanases in order to utilize needed nutrients, with no need for composting, which makes the commercial production relatively simple [26]. The substrate preparation procedure includes shredding substrate into small particle size (3 cm) and drenching it in distilled water for 24 h to reach 60% moisture content. Then, the substrate might be sterilized in polypropylene bags at 121 °C, 1.1–1.2 atm for 1–2 h [27–32] or 15 min [33,34], sterilized for 90 °C for 90 min [35], pasteurized at 60–80 °C into boiling water before the inoculation procedure [36,37], steamed over 100 °C for 4 h [38] or treated by aerobic fermentation: 65 °C for 36 h, fermentation for 12 h and aerial cooling for 24 h [39]. All other studies included substrate preparation in the same or similar manner.

After inoculation, different productivity parameters were measured during the mycelium running and harvesting period, such as mycelium running rate, time required for mycelium running completion, time required for primordia initiation and harvesting, total mushroom yield and biological efficiency [27], as well as organic matter loss, pileus, stipe diameter and stipe length [36].

### 3.2.2. Chemical Analyzes of Lignocellulosic Substrates Originated from Food Wastes

*P. ostreatus*, as a white-rot-fungi, decomposes cellulose, hemicellulose and lignin, as the main nutrient source for its mycelium growth [7]. Cellulose, hemicellulose and lignin are the major constituents of lignocellulosic wastes, which make them ideal for mushroom cultivation. The major source of cellulose is vascular plants' cell wall, and it is constructed from D-glucose units through β (1→4)-glucosidic bonds. Cellulose and hemicellulose belong to carbohydrates, and their bonds can be broken using acids or enzyme activity [40]. Additionally, the factors significant for mycelial growth, yield and efficiency of mycelium production include the range of C/N ratio, pH and moisture measurement [41]. According to Chang and Miles [42], most fungi require moisture content of the substrate between 50 and 75%, which supports maximum growth level. Carbon and nitrogen have an effect on mycelial exopolysaccharide production in submerged culture, which leads to the conclusion

that fungal cell walls can be supported by nutritional signals or environmental stress [43]. According to Choi [44], nitrogen converts to ammonia during the fermentation process, which causes an interruption of mycelial growth at high amounts. Ideal values of nitrogen are between 0.5 and 2% [45]. Nitrogen-rich compounds, used as additives to mushroom cultivation substrate, result in higher mushroom yields and increased fungal metabolic activities triggered by the presence of extra nitrogen [46,47], which contradicts the claim that nitrogen is the cause of mycelial growth interruption [44]. On the other hand, the addition of nitrogen-rich materials may also lead to higher contamination risks by competitor microorganisms [48]. Substrate supplementation is the solution of successful fermentation of different lignocellulosic substrates intended for mushroom cultivation [49].

Regarding the physical properties of the substrate, granulometric profile and particle size specifies the surface area available for mycelial growth. Smaller particles result in substrate compression and lack of gas exchange and availability of substrate molecules for the hydrolitic enzymes responsible for mycelium growth, which limits mushroom yield [50].

Prior to inoculation, analysis of olive pruning residues mostly included pH measuring, electroconductivity, total organic matter content and moisture content and C/N ratio, besides cellulose, hemicellulose and lignin content. Atomic absorption spectrophotometry was used for mineral composition analysis (K, Ca, Mg, Na, Fe and Mn) [36]. Additionally, Sakellari et al. [51] analyzed the elemental composition of substrates with different ratios of olive leaves, olive mill waste and grape marc: Al, As, Ba, Cd, Co, Cr, Cs, Cu, Fe, K, Mn, Na, Ni, Pb, Rb, Sr, V and Zn. On the other hand, olive mill waste together with grape marc as mushroom substrates, were analyzed for total phenolic content, individual phenolic compounds, terpenics and squalene [32]. In addition to major constituent analysis, Melanouri et al. [52] included total nitrogen and organic matter determination of substrates containing grape pomace, coffee residue and olive pulp. Analysis of spent coffee grounds, corn and rice residues, respectively, involved the moisture content, total C and N content and pH measurement [25,53]. Adebayo et al. [38] included cellulose, hemicellulose and lignin analysis of palm and rice wastes, while Ma et al. [54] analyzed nutrient content of substrates consisting of used diaper and food waste, banana skin, coffee waste and sugarcane bagasse, and C/N and total N content of coconut fiber, coffee husk and corn bran substrate mixtures [30]. Beer waste (spent brewery grains) was analyzed for pH and C and N determinations [34], while Rugolo et al. [55] and Fernandes Pereira et al. [56] included phosphorus, C/N ratio and humidity analysis. Sugarcane bagasse substrate analyses included cellulose, hemicellulose, lignin, total N, total C, P, Ca, Mg, Na and K [57], C/N ratio, S, H, pH and moisture content analysis [58], while rice substrate was additionally investigated for total K, P and heavy metals (Cr, As, Cd, Hg and Pb) [59]. Total N, organic matter, pH and electrical conductivity were investigated for soybean, olive and winery wastes [29,60].

### 3.2.3. Composition of Food Waste Substrates Used for Mushroom Cultivation

Olive waste substrates prepared in experiment by Koutrotsios et al. [28] contained olive pruning residues and two-phase olive mill waste (TPOMW) mixed with wheat straw in ratios 25, 50 and 75%, as well as the mixture of both wastes in ratios of 25 and 50%, while Fayssal et al. [36] used olive pruning residues only in a combination with wheat straw (1:3, 3:1). Extensive literature covers the preparation of substrate mixtures of olive and grape (wine industry) wastes. Thus, Koutrotsios et al. [32] and Sakellari et al. [51] used TPOMW, olive leaves, grape marc and wheat straw, as well as their combinations in ratios 3:1, 1:1 and 1:3. Olive and grape wastes, along with wheat straw, were used as substrates by Tagkouli et al. [61] and Tsiantas et al. [62] in the following combinations: wheat straw:grape marc (1:1), olive leaves:TPOMW (3:1) and wheat straw as control. Grape pomace as the only food waste substrate was utilized in the research of Doroški et al. [27], in the following mixtures with wheat straw: 100%, 80%:20% and 50%:50%.

Coffee waste covered in the literature was either prepared in a mixture with wheat bran and straw [52,63], while Carrasco-Cabrera et al. [25] utilized spent coffee grounds in a mixture with sawdust. Ling Ma et al. [54] included coffee and banana waste in a food waste mixture with the addition of diaper waste. Additionally, Nguyen et al. [64] used spent coffee grounds in the following formulations: 100%, in the ratios 50%:50% and 20%:80% with wheat straw and cardboard. Coffee and cocoa husk in the mixture with other wastes were used in the work of Lowor and Ofori [65]. Cocoa and palm wastes were utilized as substrates in the following studies: Mota da Silva et al. [66] mixed palm oil waste and cocoa almond peels in five substrates in different ratios; Fernandes Pereira et al. [56] used cocoa pod shells and beer waste residues, where beer spent grains varied between 10 and 90% (*w/w*), while cocoa waste was used as a supplement to the weight of 100 g. Palm oil wastes, bunches and shafts, were used in the other study [38] in combination with rice bran, sawdust and wheat bran, 100% of each substrate alone and in the combination of 50%:50% of bunch and shaft with sawdust, rice or wheat residues. Palm shell was used to produce biochar, then utilized as bio-fertilizer for oyster mushroom growth [67]. Biochar was mixed with rice bran and sawdust in three different mixtures, where biochar was weighted between 10 and 30 g, rice bran between 84 and 86 g and sawdust >850 g. On the other hand, different palm seeds in combination with shells of Brazil nuts and pine sawdust were studied as cultivation substrate by Vieira Bentolila de Aguiar et al. [68], while Zakil et al. [58] mixed empty fruit bunches, palm pressed fibers, fresh fruit bunches in combination with sugarcane bagasse and rubber tree sawdust, mostly in percentages 25%:75% and 50%:50%. Economou et al. [29] used a different approach and utilized spent mushroom substrate supplemented with wheat bran and soybean flour for a new mushroom cultivation process in order to obtain C/N ratios effective in cultivation. On the other hand, sunflower husks were used for oyster mushroom cultivation in a mixture with wheat straw (3:2) [39], while pure and degraded hazelnut husks (5%) were prepared in a mixture with the polymer matrix in the study of Duzkale Sozbir [35]. Peanut hulls and nuts mixed in different ratios (20%:80%, 50%:50%, 100%) were used in the study of Zied et al. [69] for *P. ostreatus* substrate supplementation. Beer wastes were used in the following studies: a pristine soil was mixed with previously immobilized spent brewery grains with fungal mycelium in a 1:5 ratio [34], malted barley (spent brewery grains) was used as supplementation for other agricultural waste substrates [55], while three categories of beer wastes: brewer's spent grain, hot trub and residual yeast was mixed with cocoa pod shells in different ratios [56]. Submerged liquid and solid-state fermentation were applied in the following studies: banana pseudostem and coconut fiber in the amount of 20 g were added into vials with Kirk's culture medium [31], and mycelial discs were inoculated for growth. This study covered the enzymatic activity determination. On the other hand, coconut oil cake, together with sesame oil cake, were prepared in conical flasks for mycelial inoculation in order to produce hydrophobin-like proteins [33]. Agro-cereal residues, including rice, corn and sugarcane wastes, were mostly prepared pure for fungi inoculation: rice straw and corn stubble were used in the study of Zarate-Salazar et al. [53], while Akter et al. [70] used rice husk and sugarcane bagasse additionally. Huang et al. [59] utilized rice straw, while Wiafe-Kwagyan et al. [71] additionally used rice bran and husk mixtures supplemented with different percentages of $CaCO_3$. Corn straw and sugarcane bagasse were mixed (50%:50%) in combination with plantain midrib in order to find out the best substrate for cultivation [72], while corncob alone and in the mixture with finger millet straw and bamboo waste was used in the following study [73].

### 3.3. Research Covering *P. ostreatus* on Different Substrates

3.3.1. Productivity of *P. ostreatus* Growth on Food Waste Substrates

Productivity of *P. ostreatus* growth is presented through a wide range of parameters. First of all, the studies include biological efficiency of substrate utilization (BE) as the most valuable parameter that involves the yield of the cultivated fruiting bodies per kg of substrate on a dry weight basis [27–29,32,36,37,39,52,53,55,58,59,65,66,68,70,71,73]. This

parameter is followed with mycelium running rate in bags (mm/day) [27,28,52,70], total mushroom yield (g) [27,28,32,37,54,58,65,67–69,71–73], time required for completion of mycelium running (days), time required for primordia appearance and harvesting (days) [27–29,35,37,39,54,58,60,64,70,72,73]. On the other hand, some investigations included economic yield (g/bag) after removal of inedible parts of the mushroom and organic matter loss [36,53,54,68], as well as mushroom weight (g) [29,32,36,39,70], pileus and stipe diameter (cm) and stipe length (cm) [36,39,52–54,66,70–73]. Additionally, productivity parameter defined as the ratio of BE over the duration of the crop cycle was included in the study of Koutrotsios et al. [32] and Cayetano-Catarino et al. [37], while Ling Ma et al. [54] and Zied et al. [69] included the average number of fruiting bodies and primordia. Total biomass of fresh mushrooms [53], percent of visual mycelium [70], number of clusters [66,69] and total mycelial mass [29] were rare parameters included in the calculation.

### 3.3.2. Chemical Analyzes of *P. ostreatus* Fruiting Bodies Cultivated on Food Waste Substrates

Many fungi species produce polysaccharides with very important medicinal valuation, such as anticancer and immunomodulation properties. Moreover, the nutritional composition of cultivated mushrooms strongly depends of the substrate from which it originates. With respect to the fact, many studies cover a wide range of chemical analysis of fresh and dry fruiting bodies and their extracts [27,51]. Many studies do not include nutritional and chemical analysis of fruiting bodies, only their productivity value and morphology.

Accordingly, the analysis of oyster mushroom-fruiting bodies originating from olive mill waste included ash, crude fiber and crude fat, followed by nitrogen content, crude protein, total carbohydrates and gross energy, as well as total phenolic content and antioxidant activity—the radical scavenging activity (DPPH) and ferric ion reduction power (FRAP). In addition, $\alpha$- and total glucans were determined [28].

On the other hand, Fayssal et al. [36] included moisture content, total soluble sugar, fructose, glucose and sucrose, as well as mineral composition and fatty acid profile of mushrooms. Individual phenolic compounds and terpenics were the additional part of an investigation conducted by Koutrotsios et al. [32], as well as squalene and ergosterol content. Elemental composition of mushroom-fruiting bodies cultivated on the substrates containing grape marc and olive wastes were part of the investigation conducted by Sakellari et al. [51], which calculated toxic and essential element intake through consumption, while the volatile aroma compounds profile of fruiting bodies were the aim of the investigation conducted by Tagkouli et al. [61]. Determination of ergothioneine and lovastatin content of mushrooms cultivated on grape and olive wastes in comparison to conventional substrates was investigated in the study of Tsiantas et al. [62].

Regarding grape pomace substrate, Doroški et al. [27] included spectrophotometric methods for total polyphenolics, polysaccharides and protein determination, as well as antioxidant activity—ABTS and DPPH. Overall, obtained values were converted into quality indices in order to achieve total quality of every single sample through a total quality index approach [74].

Spent coffee grounds as a substrate for *P. ostreatus* were investigated by Carrasco-Cabrera et al. [25], more precisely the extraction and quantification of caffeine and caffeine metabolites in the cultivated fruiting bodies in comparison to its concentration in the spent substrate. On the other hand, extensive chemical analysis of mushrooms grown on cocoa and coffee husk was conducted by Lowor and Ofori [65] and includes moisture content, crude protein, ethanol soluble sugars, ash, lipid and total nitrogen content and mineral nutrients.

Nutritional value of fruiting bodies cultivated on cocoa and palm waste [66] included dry matter, mineral matter, crude protein, ether extract, fiber in neutral detergent and acid detergent and lignin. Carbon, C/N ratio, nitrogen, total carbohydrates and energy calculation were also the part of the analysis. Zakil et al. [58] and de Agioar et al. [68]

analyzed protein, total fat, total carbohydrate, moisture, energy and crude fiber in the frame of their investigation regarding sugarcane and palm wastes. Regarding the nut–fruit originated wastes and beer waste, no study included any chemical analysis of the produced fruiting bodies.

Akter et al. [70] analyzed the fruiting bodies cultivated on rice, corncob and sugarcane wastes for total sugar content, protein content, ash content and total phenolics using spectrophotometric and gravimetric methods. Nutritional value including moisture, ash, crude polysaccharide, protein, fiber and fat contents of mushrooms cultivated on rice waste was determined by Huang et al. [59].

### 3.3.3. Future Challenges and Recommendations

Regarding the overall studies conducted in the last five years, a wide range of food wastes were tested as an alternative substrate for *P. ostreatus* cultivation. The analyses covered in those studies mostly included the morphology of fruiting bodies and characteristics of used substrates, lacking information on chemical and nutritional composition of fruiting bodies. The calculation of the economic profitability of alternative substrate commercial utilization may be included as a new investigation, which is of great importance for potential implementation by future manufacturers.

### 4. Conclusions

In general, sustainable food waste management is applicable in providing an initial resource for a new process which produces high-quality food from low quality waste in mushroom cultivation. This study covered three categories of food waste: food-processing wastes, agro-cereal wastes and nut–fruit wastes, which were used the most for the cultivation of *P. ostreatus* during 2017–2022. According to the overall results, the analyses that are most covered concern the productivity of the substrate, the morphological characteristics of the fruiting bodies cultivated on a specific substrate and the chemical characterization of the used substrate and substrate depletion after cultivation. Certain studies covered submerged liquid and solid state fermentation related to the enzymatic activity determination. Chemical analyses of mushroom-fruiting bodies cultivated on food waste are lacking, considering the fact that chemical composition of cultivated fruiting bodies may be related to the substrate from which it originates, which provides the opportunity for additional research in the future investigations.

**Author Contributions:** Conceptualization, A.D. and I.D.; methodology, A.D. and I.D.; formal analysis, All authors; investigation, All authors; writing—original draft preparation, A.D.; writing—review and editing, all authors. All authors have read and agreed to the published version of the manuscript.

**Funding:** This research is supported according to the agreement on the realization and financing research work in 2022 between the Faculty of Agriculture in Belgrade and the Government of the Republic of Serbia, Ministry of Education, Science and Technical Development, record number 451-03-68/2022-14/200116.

**Institutional Review Board Statement:** Not applicable.

**Informed Consent Statement:** Not applicable.

**Data Availability Statement:** Not applicable.

**Conflicts of Interest:** The authors declare that they have no known competing financial interest or personal relationships that could have appeared to influence the work reported in this paper.

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
