# Peer review of "Food Waste Originated Material as an Alternative Substrate Used for the Cultivation of Oyster Mushroom (Pleurotus ostreatus): A Review"

_sustainability, doi:10.3390/su141912509_

Round 1

Reviewer 1 Report

Dear authors,

the present manuscript is an interesting review.

Some minor recommandations:

- please check the references in the text, e.g.

lines 46-48: references 21 and 24 cited?! maybe typing mistake

lines 186-207 - same please correct

I would suggest to add a new section future perspectives and recommandations, a practical checklist of available intrinsic and extrinsic factors.

Maybe already in the introduction a short paragraph about these factors.

Author Response

Reviewer 1

Dear authors,

the present manuscript is an interesting review.

Thank you very much for your objective opinion.

Some minor recommandations:

- please check the references in the text, e.g.

lines 46-48: references 21 and 24 cited?! maybe typing mistake

lines 186-207 - same please correct

Thank you for this suggestion: we checked all manuscript references and corrected the mistaken order formed automatically by the reference manager.

I would suggest to add a new section future perspectives and recommandations, a practical checklist of available intrinsic and extrinsic factors.

Maybe already in the introduction a short paragraph about these factors.

Authors added new section in the end of the manuscript named „Future challenges and recommendations“.

Reviewer 2 Report

In my opinion, the manuscript entitled Food waste originated material as an alternative substrate used for the cultivation of oyster mushroom (Pleurotus ostreatus): a review by Doroški et al., is a very good one being an up-to-date review regarding the use of food waste in Pleurotus ostreatus cultivation. This is an important review that explore the idea of the reuse of food in different sectors.

I have only some small comments, as follows:

1. When authors first mentioned the abbreviation of Pleutorus ostreatus in line 17, it should be Pleutorus ostreatus (P. ostreatus), or authors could directly mention the abbreviation in line 10.

2. The introduction could be improved by introducing new articles regarding the chemical composition of Pleurotus species, their taxonomy and botanical description, their worldwide consumption, and cultivation method. For instance, authors could include next recent published articles:

·         Green Biotechnology of Oyster Mushroom (Pleurotus ostreatus L.): A Sustainable Strategy for Myco-Remediation and Bio-Fermentation, https://doi.org/10.3390/su14063667

·         Cultivation of Pleurotus ostreatus https://doi.org/10.1002/9781119149446.ch16

·         Green potential of Pleurotus spp. in biotechnology, 10.7717/peerj.6664

3. Also, regarding the sustainability of Pleutorus spp. cultivation, authors should go further and mentioned also some recently published articles such as:

·         Use of green light to improve the production of lignocellulose-decay enzymes by Pleurotus spp. in liquid cultivation, https://doi.org/10.1016/j.enzmictec.2021.109860

4. In introduction, I think that authors should check the line spacing option for paragraphs.

5. Line 85: I think the word in is repeated twice. Please remove one word.

6. Line 169: there is an extra space between content.   Then,….. please remove it.

Thank you!

Author Response

Reviewer 2

In my opinion, the manuscript entitled Food waste originated material as an alternative substrate used for the cultivation of oyster mushroom (Pleurotus ostreatus): a review by Doroški et al., is a very good one being an up-to-date review regarding the use of food waste in Pleurotus ostreatus cultivation. This is an important review that explore the idea of the reuse of food in different sectors.

Thank you very much for your objective opinion.

I have only some small comments, as follows:

  1. When authors first mentioned the abbreviation of Pleutorus ostreatusin line 17, it should be Pleutorus ostreatus (P. ostreatus), or authors could directly mention the abbreviation in line 10.

Thank you for this suggestion. The authors added the abbrevation in the abstract (line 17).

  1. The introduction could be improved by introducing new articles regarding the chemical composition of Pleurotus species, their taxonomy and botanical description, their worldwide consumption, and cultivation method. For instance, authors could include next recent published articles:
  • Green Biotechnology of Oyster Mushroom (Pleurotus ostreatusL.): A Sustainable Strategy for Myco-Remediation and Bio-Fermentation, https://doi.org/10.3390/su14063667
  • Cultivation of Pleurotus ostreatushttps://doi.org/10.1002/9781119149446.ch16
  • Green potential of Pleurotusspp. in biotechnology, 10.7717/peerj.6664
  1. Also, regarding the sustainability of Pleutorusspp. cultivation, authors should go further and mentioned also some recently published articles such as:
  • Use of green light to improve the production of lignocellulose-decay enzymes by Pleurotusspp. in liquid cultivation, https://doi.org/10.1016/j.enzmictec.2021.109860

Thank you. We improved and expanded introduction section with listed references.

  1. In introduction, I think that authors should check the line spacing option for paragraphs.

We checked line spacing and adapted it to the rest of the manuscript.

  1. Line 85: I think the word in is repeated twice. Please remove one word.

Thank you, we deleted redundant word.

  1. Line 169: there is an extra space between content.   Then,….. please remove it.

We removed extra space you suggested.

Thank you!
